# Clinical manifestations, prevalence, risk factors, outcomes, transmission, diagnosis and treatment of COVID-19 in pregnancy and postpartum: a living systematic review protocol

Magnus Yap,[1] Luke Debenham,[1] Tania Kew,[1] Shaunak Rhiju Chatterjee [ID] ,[1] John Allotey,[2,3] Elena Stallings,[4,5] Dyuti Coomar,[3] Siang Ing Lee,[3] Xiu Qiu,[6,7,8] Mingyang Yuan,[3,7] Anna Clavé Llavall,[1] Anushka Dixit,[1] Dengyi Zhou,[1] Rishab Balaji,[1] Madelon van Wely,[9] Elena Kostova,[9] Elisabeth van Leeuwen,[10] Lynne Mofenson,[11] Heinke Kunst,[12] Asma Khalil,[13] Simon Tiberi,[12,14] James Thomas,[15] Vanessa Brizuela [ID] ,[16] Nathalie Broutet,[16] Edna Kara,[16] Caron Kim,[16] Anna Thorson,[16] Pura Rayco-Solon,[16] Hector Pardo-Hernandez,[16] Olufemi Taiwo Oladapo,[16] Javier Zamora [ID] ,[17] Mercedes Bonet,[16] Shakila Thangaratinam,[2,18] On behalf of PregCOV-19 Consortium

MY, LD, TK and SRC are joint first authors.

For numbered affiliations see end of article.

**Correspondence to**
Dr John Allotey;
j.allotey.1@bham.ac.uk

## ABSTRACT

**Introduction** Rapid, robust and continually updated evidence synthesis is required to inform management of COVID-19 in pregnant and postpartum women and to keep pace with the emerging evidence during the pandemic.
**Methods and analysis** We plan to undertake a living systematic review to assess the prevalence, clinical manifestations, risk factors, rates of maternal and perinatal complications, potential for mother-to-child transmission, accuracy of diagnostic tests and effectiveness of treatment for COVID-19 in pregnant and postpartum women (including after miscarriage or abortion). We will search Medline, Embase, WHO COVID-19 database, preprint servers, the China National Knowledge Infrastructure system and Wanfang databases from 1 December 2019. We will supplement our search with studies mapped by Cochrane Fertility and Gynaecology group, Evidence for Policy and Practice Information and Co-ordinating Centre (EPPI-Centre), COVID-19 study repositories, reference lists and social media blogs. The search will be updated every week and not be restricted by language. We will include observational cohort (≥10 participants) and randomised studies reporting on prevalence of COVID-19 in pregnant and postpartum women, the rates of clinical manifestations and outcomes, risk factors in pregnant and postpartum women alone or in comparison with non-pregnant women with COVID-19 or pregnant women without COVID-19 and studies on tests and treatments for COVID-19. We will additionally include case reports and series with evidence on mother-to-child transmission of SARS-CoV-2 in utero, intrapartum or postpartum. We will appraise the quality of the included studies using appropriate tools to assess the risk of bias. At least two independent reviewers will undertake study selection, quality assessment and data extraction every 2 weeks.

## Strengths and limitations of this study

► Our living systematic review will be underpinned by a comprehensive literature search, study quality assessment and appropriate planned meta-analysis to efficiently collate the overall findings on COVID-19 in pregnant and postpartum women.
► We will continuously update our search, study selection, data extraction, analysis and reporting of the findings at prespecified time intervals.
► Rapid publication of new studies with new outcomes of interest, screening and testing strategies and reporting of novel treatments may require changes in the review protocol, specifically regarding data extraction and search strategies for future updates.
► This review may be subjected to publication bias, since studies with perceived positive results may be published faster than those with perceived negative results, thus we will continuously review registries of randomised controlled trials to detect trials that should have reported results, but that have not done so and will contact corresponding authors to obtain results.
► The dynamic nature of the living systematic review requires dedicated team of committed researchers and resources, efficient peer review and support of the journal editors, to ensure that findings are rapidly published in the public domain with provision for continuous updates to inform living guidelines and policies.

We will synthesise the findings using quantitative random effects meta-analysis and report OR or proportions with 95% CIs and prediction intervals. Case reports and series will be reported as qualitative narrative synthesis. Heterogeneity will be reported as $I^2$ and $\tau^2$ statistics.

**Ethics and dissemination** Ethical approval is not required as this is a synthesis of primary data. Regular updates of the results will be published on a dedicated website (https://www.birmingham.ac.uk/research/who-collaborating-centre/pregcov/index.aspx) and disseminated through publications, social media and webinars.
**PROSPERO registration number** CRD42020178076.

## INTRODUCTION

Coronavirus disease caused by the SARS-CoV-2 was declared a pandemic by the WHO on 11 May 2020.[1] In the first 3 months of the pandemic alone, over 4.5 million individuals have been affected—the infection continues to spread rapidly.[2] Case fatality rates range from 0.4% to 3.6% depending on the country and detection method.[3 4] In previous serious coronavirus outbreaks caused by SARS and middle east respiratory syndrome, the rates of intensive care unit admission and mortality were significantly higher in infected pregnant than non-pregnant women, and adverse pregnancy outcomes were common.[5] There are concerns about the potential effects of SARS-CoV-2 infection on mothers and babies, including the risks of transmission to the fetus and neonate.[6] Some countries such as the UK have classed pregnant women as a vulnerable group requiring shielding during the pandemic.[7] Black and ethnic minority individuals are considered to be more likely to be infected and have severe COVID-19 disease than Caucasian individuals.[8–10]

There has been a rapid increase in the numbers of published studies and reports on the prevalence of SARS-CoV-2 infection in pregnancy, risk factors, mother-to-child transmission, effects on pregnant and recently pregnant women, including those who have delivered or had a recent abortion, and their babies.[11] Traditional systematic reviews are not able to keep pace with the rapid pace of publications and quickly become outdated.[12] Furthermore, numerous systematic reviews, addressing similar questions and including identical numbers of studies that differ very slightly from each other, make it challenging for guideline makers to identify the up-to-date evidence.[13 14] Many of these reviews do not follow reporting guidelines and other general principles of conducting robust systematic reviews, often including case series and case–control studies in the meta-analysis resulting in biassed estimates.

Any recommendation on the care of pregnant women and recently pregnant women with suspected or confirmed COVID-19, and their babies, should be based on robust evidence. Clinicians need a single point of reference that comprehensively provides up-to-date evidence for key questions. In a rapidly changing research and clinical environment, this requires a clear prospective plan to update the available evidence beyond conventional systematic reviews and meta-analyses. We propose to undertake a living systematic review to address the key research questions on SARS-CoV-2 infection in pregnant and postpartum women, including after childbirth and early pregnancy loss.

## Aim

Our goal is to provide up-to date evidence on the risks and risk factors for COVID-19 and associated complications in pregnant and postpartum women and their babies through a living systematic review and meta-analysis and to assess the accuracy of diagnostic tests and effectiveness of treatment in the management of the disease.

## Objectives

In pregnant and postpartum women (including after miscarriage or abortion):
► To determine the prevalence of and risk factors for SARS-CoV-2 infection and severe COVID-19.
► To evaluate the accuracy of tests and prediction models for screening, diagnosis and prediction of COVID-19 and its complications.
► To assess the effects of interventions for prevention of SARS-CoV-2 infection.

In pregnant and postpartum women (including after miscarriage) with suspected or confirmed COVID-19:
► To study the rates of clinical symptoms and signs, laboratory and radiological manifestations of the disease and compare against non-pregnant women.
► To assess the rates and risk factors for COVID-related and pregnancy-related maternal and perinatal outcomes and compare against non-pregnant women with COVID-19 and pregnant women without the disease, respectively.
► To determine the risks and risk factors for mother–child transmission of SARS-CoV-2 in utero, intra and peripartum, the prevalence and persistence of the viral particles or immunological response in breast milk, amniotic fluid, cord blood, placenta, vaginal fluids and faeces in women and their babies (nasopharyngeal/throat swabs, blood, saliva, faeces).
► To assess the effects of interventions to prevent COVID-related complications.
► To determine whether the rates of prevalence, clinical manifestations, outcomes and risk factors vary by: (a) screening and testing strategy for SARS-CoV-2, (b) selection of populations, (c) risk status of the included women, timing of exposure (first, second, third trimester, postpartum), World Bank economic region (low, middle and high income) and quality of studies (low, high).

## METHODS

Our systematic review protocol is registered in Prospero.[15] We will regularly repeat the searches, data extraction and analyses as described in figure 1.

## Literature search

We will carry out a systematic search on the WHO Database of publications on COVID-19, the Evidence for Policy and Practice Information and Co-ordinating Centre (EPPI-Centre) map of the current evidence on COVID-19, Cochrane databases, China National Knowledge Infrastructure, Wanfang and preprint databases (ArXiv,

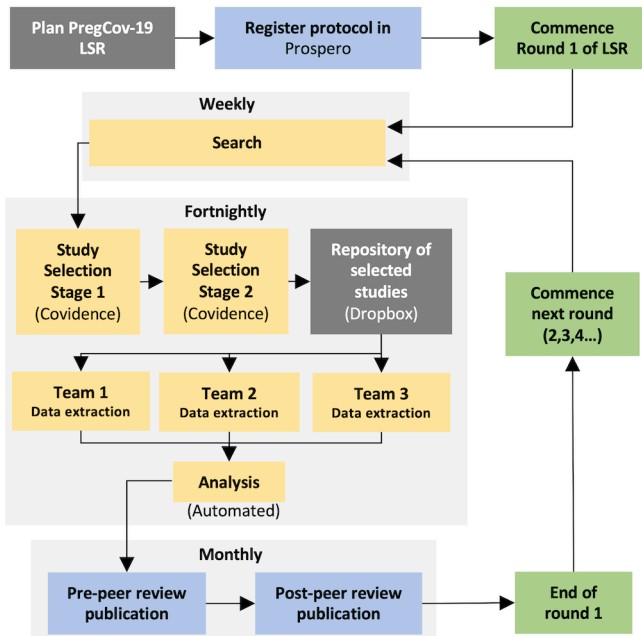

**Figure 1** Steps in the living systematic review (LSR) on COVID-19 in pregnant and postpartum women.

BiorXiv, medRxiv, search.bioPreprint), the reference lists of included studies, relevant systematic reviews and guidelines published by national and international professional societies, COVID-19 research websites[16 17] and follow blogs dedicated to the identification of primary case reports, case series, observational studies or randomised controlled trials (RCT) describing women affected by COVID-19 in pregnancy. We will also link with established groups conducting surveillance and research studies in pregnant women with COVID-19 to access their aggregate study data.[18–20] There will be no language restrictions. The search findings will be exported to Covidence (http://covidence.org/), an online programme that facilitates study selection and screening, recommended by the Cochrane Collaboration.[21] Our search will be updated weekly, and we shall review this frequency every 2 months. The search syntax for the Pubmed database is provided in online supplemental appendix 1

### Study selection, quality assessment and data extraction

Study selection will be a two-stage process: titles and/or abstracts of studies will be screened first, followed by evaluation of full texts for eligibility by two independent reviewers. Disagreements will be resolved through discussion or by consulting a third reviewer. We will include all cohort studies (with at least 10 participants) and randomised trials reporting on clinical outcomes relating to SARS-CoV-2 infection involving pregnant and postpartum (which includes postmiscarriage/abortion period) women and their babies. We defined cohort studies as those that included women based on exposure, followed-up over specified time and reported outcomes.[22] Case reports and case series will only be included to

answer the research questions relating to mother-to-child transmission.

We will look for serological (IgM) and/or reverse transcriptase PCR (RT-PCR) confirmation of infection in amniotic fluid, placenta, cord, newborn and maternal blood and newborn and maternal respiratory secretions (at birth, 24 and ≥48 hours after birth) to distinguish between congenital, intrapartum and postpartum transmission. Women with a respiratory sample (naso/oropharyngeal swab or tracheal/bronchoalveolar lavage) positive for SARS-CoV-2 RT-PCR or positive SARS-CoV-2 serological tests will be considered to have confirmed COVID-19. We will also assess risk factors such as mode of delivery, maternal disease severity, gestational age of maternal infection, preterm delivery, rooming-in and breast feeding on mother-to-child transmission. Women with a clinical diagnosis based on chest CT or radiograph or other features will be considered as suspected COVID-19. Table 1 provides the details of risk factors, maternal and perinatal outcomes that will be evaluated in the review.

We will assess the risk of bias for included cohort studies using the Newcastle Ottawa Scale for comparative cohorts,[23] the Cochrane Risk of Bias 2 tool for randomised trials,[24] Quality Assessment of Diagnostic Accuracy Studies 2 (QUADAS-2) for diagnostic accuracy studies[25] and the tool outlined by Hoy et al[26] for prevalence studies. A preplloted form will be used for data extraction of the included studies. Two reviewers will independently extract data and disagreements will be resolved by consensus or by consulting a third reviewer. We will assess for duplication of the data by comparing characteristics of the mother or baby and the settings of the studies. If required, we will contact the authors of primary studies for clarification about duplicate data. Where possible, we have planned a semiautomated process to identify duplicates and facilitate rapid update of search and data extraction.

### Analysis

We will undertake narrative syntheses and perform aggregate meta-analyses when there are at least two studies with minimal clinical heterogeneity. Dichotomous outcomes will be summarised as proportions, OR and continuous outcomes as standardised mean differences. We will use random effects model for the analysis when the number of studies permits to estimate between-study variance. To summarise proportions, we will use Freeman-Tukey transformation to stabilise variances while dealing with studies with zero events. We will provide 95% CIs and predictive intervals (PI) to report on the precision of estimates and to aid the interpretation of heterogeneity. Heterogeneity will be reported as $I^2$ and $\tau^2$ statistics.

Preplanned subgroup analysis will be by (a) suspected/probable or confirmed COVID-19, (b) diagnosis in pregnancy or postnatal period, (c) trimester of diagnosis (first, second or third), (d) country income-level (high-income or low-income and middle-income country), (e) screening strategies (universal, symptom-based or

**Table 1** Study participants, risk factors and outcomes evaluated in the living systematic review on COVID-19 in pregnant and postpartum women

| | |
|---|---|
| Population | Pregnant/ postpartum/postabortal women with suspected or confirmed COVID-19 infection |
| Risk factors | Maternal<br>Age, ethnicity, pre-existing medical conditions (including diabetes, chronic hypertension, asthma and Chronic Obstructive Pulmonary Disease), smoking, immunosuppression, gestational diabetes, symptoms, hypertensive disorders in pregnancy (pre-eclampsia, pregnancy-induced hypertension), body mass index ≥30, multiple pregnancy, in vitro fertilisation, parity, gestational age, mode of delivery, pregnancy status (pregnant or delivered), reproductive tract infections, symptoms and abnormal lab results. |
| Clinical manifestations | Symptoms and signs<br>Cough, fever, breathlessness, sputum, myalgia, fatigue, diarrhoea, headache, sore throat, chest pain, rigour, ageusia, anosmia, nausea or vomiting, Sequential Organ Failure Assessment (SOFA), Quick SOFA, asymptomatic presentation<br>Laboratory<br>White cell count, lymphocyte count, haemoglobin, anaemia, platelet count, albumin, Alanine aminotransferase, Aspartate transaminase, C-reactive protein, creatinine, lactate dehydrogenase, creatine kinase, high-sensitivity cardiac troponin, prothrombin time, D-dimer, serum, ferritin, interleukin-6, procalcitonin<br>Radiological<br>Consolidation, ground-glass opacity, bilateral pulmonary infiltration, unilateral pulmonary infiltration, abnormal chest X-ray, abnormal chest CT |
| Outcomes | Maternal COVID-related outcomes<br>Mortality: all-cause mortality, COVID-specific mortality<br>clinical respiratory syndrome: pneumonia, respiratory failure, acute respiratory distress syndrome, severe pneumonia; invasive ventilation, non-invasive ventilation, oxygenation; long-term respiratory outcomes<br>Time from illness onset to outcome (death, recovery)<br>Hospitalisation: admission to intensive care unit (ICU), admission to hospital, ICU length of stay<br>Organ failure: sepsis, septic shock, cardiac failure, coagulopathy, thromboembolism, acute cardiac injury, acute kidney injury, acute hepatic failure, cytokine storm syndrome (haemophagocytic lymphohistiocytosis), hypoproteinaemia, acidosis, central nervous system manifestations, secondary infection, duration of viral shedding<br>delirium, acute neuropsychiatric emergency, agitation, anxiety, depression, psychosis<br>Pregnancy-related outcomes<br>Preterm delivery (<37 w, spontaneous preterm delivery, induced preterm birth), preterm–premature rupture of membranes, prelabour rupture of membranes at (or near) term, miscarriage (spontaneous), induced abortion, mode of delivery, induction of labour, chorioamnionitis, wound infection, pregnancy-induced hypertension, gestational diabetes, antepartum haemorrhage, postpartum haemorrhage<br>Offspring outcomes<br>Stillbirth, neonatal death (early, late), foetal distress, foetal growth restriction postinfection, Apgar score at 1', 5'; cord blood pH, gestational age at delivery, birth weight, large-for-gestational age, small-for-gestational age, congenital malformation, hypoxic ischaemic encephalopathy, neonatal seizures, neonatal infection (other than COVID-19), neonatal sepsis, neonatal asphyxia, disseminated intravascular coagulation, necrotising enterocolitis, respiratory distress syndrome, admission to the neonatal unit, length of stay in neonatal unit<br>Mother-to-child transmission outcomes<br>Evidence of virus in amniotic fluid, cord blood, placenta, placental membranes, vaginal fluid, breast milk, neonatal throat swabs, maternal and neonatal faeces and saliva samples; IgM antibodies in cord blood, neonatal blood<br>Duration of viral shedding after COVID-19 symptoms onset and after clinical resolution of signs/symptoms in mother and in newborn |

risk-based testing), (f) maternal risk status (low or high) and (g) study quality (low or high). We will undertake additional sensitivity analysis to explore the effects of different populations (unselected, selected) and by excluding women with suspected COVID-19 and studies at high risk of bias. We will use trial sequential analysis to control for type I and II errors while accounting for updating estimations of between-study heterogeneity. A priori assumptions on statistical power, minimal clinically relevant effect and heterogeneity between trials will be used to define maximum sample size to detect such an effect. Boundaries of statistical significance according to sample size will be defined and used to determine statistical significance for each systematic review update result.

We have established a pool of peer reviewers to rapidly assess the findings. The initial prepeer-reviewed findings will be published in a dedicated website, followed by the full findings when peer review is complete. We will simultaneously submit our work for publication in scientific journals with clear reference to the version of the living systematic review provided in that submission. Where journals allow, we will update our findings in the journals at set intervals required by the journal.

## Patient and public involvement

Katie's team patients and public involvement group—a dedicated women's research and health advisory group were involved in the design of the protocol and will contribute to the interpretation and dissemination of the result.

## DISCUSSION

Our living systematic review will address key research questions relevant to SARS-CoV-2 infection in pregnancy and postpartum period. Our review is based on a prospective protocol with plans to continuously update all review processes from search to publication at specific time points. The findings of the review will directly inform living guidelines and policies as new evidence emerge.

Our detailed literature search is a major strength of the review. In addition to the WHO COVID-19 database, our links with Cochrane Fertility and Gynaecology and EPPI Centre groups and additional searches for studies mapped by dedicated websites as well as blogs and social media networks means that the chance of missing relevant studies is reduced. Through our networks with key collaborators in the WHO (maternal, neonatal, child and adolescent health) COVID-19 research group[27] and relevant working groups, we will be able to access and include unpublished data. Our collaborative links with researchers in China provide access to Chinese language databases, so that these studies are not missed.

We have developed the protocol to ensure that all stages of the review are robust by adhering to recommended methods for conduct and reporting of systematic reviews.[28] By restricting case reports and series to the research question on mother-to-child transmission alone, we will not only ensure that relevant cases are not missed but also avoid biased estimates for other research questions on risk factors and prevalence. Determination of mother-to-child SARS-CoV-2 transmission is particularly challenging since there is no consensus on what constitutes intrauterine, intrapartum and postpartum transmission (including breast milk vs horizontal transmission).[6] To maximise data collection addressing this question, we will include case reports and case series for this part of the review. We will collect data on all types of samples to enable evaluation of varying definitions of the evidence required for confirmation of mother-to-child SARS-CoV-2 transmission and timing.

Our work is subjected to some limitations. Our protocol has been developed based on our current knowledge of the COVID-19. As the pandemic unfolds, emerging new evidence may require changes in the protocol. The systematic review will be influenced by the characteristics of the individual studies, which may comprise heterogeneous populations, definitions of COVID-19, sampling frames, test strategies for diagnosis, definitions and reporting of outcomes. We have addressed these challenges by clearly defining the inclusion criteria and by exploring heterogeneity through sensitivity and subgroup analyses. The findings will be reported as 95% CI to communicate the uncertainty around the pooled estimates and PI to anticipate the variability in new study estimates. Given the urgency of the situation in the pandemic era, many studies are published as preprints, often followed by a full publication at a later date. The living systematic review needs to be responsive to any changes in data between the preprint to full publication stage. Furthermore, there is a risk of duplicate data, as individual studies may report the findings of the same mother–baby data that have been published elsewhere. We plan to exclude studies with suspected duplicate data and contact the authors of the primary studies for clarification. This review may be subjected to publication bias, since studies with perceived positive results may be published faster than those with perceived negative results. We will continuously review registries of RCTs to detect trials that should have reported results but that have not done so and will contact corresponding authors to obtain results.

Many of the automated tools for living systematic review are mainly developed for reviews on RCTs and not for observational studies.[29] Unlike traditional systematic reviews, living systematic reviews require substantial investment in time and human resources and resources to regularly update the findings. Sustained funding is required to maintain the same level of output over a longer period of time. This is particularly challenging, as the publication rate of studies on SARS-Cov-2 infection in pregnancy and postpartum is likely to increase exponentially. To ensure that all relevant studies are identified in a timely manner, in addition to traditional searches, we will use our collaborations with other global efforts to map studies on COVID-19 and pregnancy and postpartum and automated alerts to identify new evidence when it gets published. Given the wide scope of this review, numerous reviewers will be involved, requiring clear operating procedures and pathways in place for workflow, training, monitoring and quality assessment. The necessity to swiftly publish the collated evidence needs to be balanced against publishing the findings after peer review. To be able to sustain the level of reviewer turn-around every 2 weeks will be a challenge.

To date, apart from Cochrane, very few journals provide specific guidelines for publication of living systematic reviews and its subsequent update.[30] Various models of publication have been considered.[31] The first model is similar to the Cochrane reviews, where with each new

 

update (usually done in yearly intervals), there is a new publication with a new version and DOI in PubMed with linked updates between versions. This model can also take into account any changes in authors between the versions.[32] However, in the current situation of rapidly evolving evidence, this is likely to result in numerous publications, even if reported on a monthly basis. In the second model, the introduction and methods of the main manuscript do not change, with only changes in the results section, which is written in such a way that most of the information is provided in tables and figures that are revised along with the abstract. The Discussion section of the newer version can incorporate a paragraph on the implications of new findings. The manuscript will be less resource intensive to prepare in the second model, but the original version should have been written in a generic manner to accommodate new information emerging in subsequent versions. In another model, the findings of subsequent updated analyses appear as new appendices, with no changes in the original abstract or manuscript. While this model requires less efforts from authors, editors and peer reviewers, it can mistakenly provide inaccurate old evidence if readers only access the abstract. Furthermore, this model makes it difficult to add or remove authors according to the changes in their contribution to subsequent versions.

In the midst of the current pandemic, much still remains unknown about how SARS-CoV-2 infection affects pregnant and postpartum women and their babies compared with reproductive aged non-pregnant women. It is essential that clinicians' decisions to manage pregnant and postpartum women and their babies with COVID-19 are guided by the evidence. Our living systematic review is well suited to rapidly provide updated findings for translation into clinical practice. The flexibility of the living systematic review needs to be matched by a willingness of journal editors and guideline makers to provide a framework that allows rapid dissemination of the new findings.

## Ethics and dissemination
No ethical concerns. The findings will be disseminated through a designated website, publications, presentations in webinars and social media.

### Author affiliations
[1]Birmingham Medical School, College Medical and Dental Sciences, University of Birmingham, Birmingham, UK
[2]WHO Collaborating Centre for Global Women's Health, Institute of Metabolism and Systems Research, University of Birmingham, Birmingham, UK
[3]Institute of Applied Health Research, University of Birmingham, Birmingham, UK
[4]Clinical Biostatistics Unit, Hospital Universitario Ramón y Cajal (IRYCIS), Madrid, Spain
[5]CIBER Epidemiology and Public Health (CIBERESP), Madrid, Spain
[6]Department of Woman and Child Health Care, Guangzhou Women and Children's Medical Center, Guangzhou Medical University, Guangzhou, China
[7]Division of Birth Cohort Study, Guangzhou Women and Children's Medical Center, Guangzhou Medical University, Guangzhou, China
[8]Department of Obstetrics and Gynecology, Guangzhou Women and Children's Medical Center, Guangzhou Medical University, Guangzhou, China
[9]Netherlands Satellite of the Cochrane Gynaecology and Fertility Group, Amsterdam University Medical Center, Amsterdam, The Netherlands
[10]Department of Obstetrics and Gynaecology, Amsterdam University Medical Center, Amsterdam, The Netherlands
[11]Elizabeth Glaser Pediatric AIDS Foundation, Washington DC, Maryland, USA
[12]Blizard Institute, Queen Mary University of London, London, UK
[13]Department of Obstetrics and Gynaecology, St George's University London, London, UK
[14]Department of Infectious Diseases, Barts Health NHS Trust, London, UK
[15]UCL Institute of Education, University College London, London, UK
[16]Department of Sexual and Reproductive Health and Research, UNDP/UNFPA/UNICEF/WHO/World Bank Special Programme of Research, Development and Research Training in Human Reproduction (HRP), World Health Organization, Geneva, Switzerland
[17]Women's Health Research Unit, Queen Mary University of London, London, UK
[18]Department of Obstetrics and Gynaecology, Birmingham Women's and Children's NHS Foundation Trust, Birmingham, UK

**Contributors** All authors contributed to the development of the protocol and writing of the manuscript. MY, LD, TK, SRC—wrote the first draft of the manuscript. JA, ES—designed the study. DC, SIL—reviewed the manuscript and were involved in quality assessment and data extraction. They have approved the final version of the manuscript. ACL, AD, DZ, RB—reviewed the manuscript and are involved in study selection, and they have approved the final version of the manuscript. XQ, MY, MvW, EK, EvL, LM, HK, AK, ST, JT, VB, NB, EK, CK, AT, PR-S, HP-H, OTO—reviewed the manuscript, contributing critical changes, and they have approved the final version of the manuscript. JZ reviewed the manuscript and will conduct the statistical analysis. He approved the final version of the manuscript. MB designed the study, and reviewed the manuscript and has approved the final version of the manuscript. ST conceived and designed the study and reviewed the manuscript and approved the final version of the manuscript. ST is the guarantor. All authors reviewed the manuscript, contributing critical changes and approved the final version of the manuscript.

**Funding** The work is partially funded by the WHO. The authors alone are responsible for the views expressed in this manuscript and they do not necessarily represent the decisions, policy or views of the funding bodies. The views of the funding bodies have not influenced the content of this manuscript.

**Competing interests** None declared.

**Patient consent for publication** Not required.

**Provenance and peer review** Not commissioned; externally peer reviewed.

**ORCID iDs**
Shaunak Rhiju Chatterjee http://orcid.org/0000-0002-3444-3948
Vanessa Brizuela http://orcid.org/0000-0002-4860-0828
Javier Zamora http://orcid.org/0000-0003-4901-588X

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
