## [Reviewer comments · BMJ Open]

ARTICLE DETAILS

TITLE (PROVISIONAL)	Clinical manifestations, prevalence, risk factors, outcomes, transmission, diagnosis and treatment of coronavirus disease (COVID-19) in pregnancy and postpartum: A living systematic review protocol
AUTHORS	Yap, Magnus; Debenham, Luke; Kew, Tania; Chatterjee, Shaunak; Allotey, John; Stallings, Elena; Coomar, Dyuti; Lee, Siang; Qiu, xiu; Yuan, Mingyang; Clavé Llavall, Anna; Dixit, Anushka; Zhou, Dengyi; Balaji, Rishab; van Wely, Madelon; Kostova, Elena; van Leeuwen, Elisabeth; Mofenson, Lynne; Kunst, Heinke; Khalil, Asma; Tiberi, Simon; Thomas, James; Brizuela, Vanessa; Broutet, Nathalie; Kara, Edna; Kim, Caron; Thorson, Anna; Rayco-Solon, Pura; Pardo-Hernandez, Hector; Oladapo, Olufemi; Zamora, Javier; Bonet, Mercedes; Thangaratinam, Shakila

VERSION 1 – REVIEW

REVIEWER	Yoshitsugu Chigusa Graduate School of Medicine, Kyoto University, Japan
REVIEW RETURNED	18-Jul-2020

GENERAL COMMENTS	It is an urgent task to elucidate the effects of coronavirus infection on pregnant and postpartum women while the end of coronavirus infection cannot be seen. The studies planned by the authors are clear and sound in purpose and method and beneficial to many clinicians. Reviewer believe that this work should be carried out promptly.
--

REVIEWER	Mohd Zulfakar Mazlan Universiti Sains Malaysia Malaysia
REVIEW RETURNED	19-Jul-2020

GENERAL COMMENTS	why don't you include study which is registered in clinical trial as well. Some of the study results there is not published. Please include web of science database as well. Suggest to register the study in US clinical trial Suggest to do flow chart Suggest to determine sample size for each objectives based on previous literature
--

REVIEWER	Liona Poon
-----------------	------------

	The Chinese University of Hong Kong Hong Kong SAR
REVIEW RETURNED	21-Jul-2020

GENERAL COMMENTS	Thank you for asking me to review this manuscript, which describes the protocol of an important study that aims to address many questions relating to the impact of SARS-CoV-2 infection on maternal and perinatal outcomes. I have the following comments / questions prior to acceptance: The schedule described is as follows:  1. Weekly search 2. 2-weekly quality assessment and data extraction 3. Monthly updates on results Can the authors elaborate on the rationale behind the proposed schedule? For the monthly updates, is there a cut off date? Are you reporting on analysis from 2 weeks before? Will there be a final publication? I believe the aim to evaluate mother-child transmission is an ambitious one as most data provide insufficient evidence for in utero, intra and peripartum mother-child transmission. Most publications have focused on early neonatal SARS-CoV-2 infection. You may wish to revise this aim to allow some flexibility. Will you include preprint papers or published papers only? What does "our search will be updated weekly" mean? Does it mean you will run the search weekly? It is unclear how you will include nonpregnant cases with SARS-CoV-2 infection. Please explain how you will capture such data systematically. Table 1: Risk factors - consider adding respiratory disease, cardiovascular disease, immunosuppression, smoking. Parity, gestational age, mode of delivery, pregnancy status should not be considered as risk factors. Outcomes - consider adding thromboembolism Offspring outcomes - there was a recent case of neonatal white matter injury; will you have "others"? Regarding duplicates, it is good that you have briefly mentioned that you have a plan to address this issue. Please be mindful that the same cases can be published by obstetricians, neonatologists, radiologists, intensivists, pathologists, admitting physicians, receiving physicians, etc. This is a potential problem that would either over- or under-estimate the effect of SARS-CoV-2 infection on maternal and perinatal outcomes. You have also mentioned that you plan to have a semi-automated process to identify these duplicates. It would be good to know more about this semi-automated process as supplementary materials. Please explain how you will handle missing data. I believe this could also be a significant issue, especially in regard to the earlier data. Such data are important as they represent the impact of the disease during the beginning of the outbreak. I also assume that the earlier data would be of low quality. One of the objectives is to
---

	determine if the rates of prevalence, clinical manifestations, outcomes and risk factors vary by: a) screening and testing strategy for SARS-CoV-2, b) selection of populations, c) risk status of the included women, timing of exposure (first, second, third trimester, postpartum), World Bank economic region (low, middle and high income) and quality of studies (low, high). In your pre-planned subgroup analysis, quality of studies was not included. In addition, with your objective / pre-planned subgroup analysis, there was an item called maternal risk status. What does this item mean? Please describe the process of establishing a pool of peer reviewers. It would be good to know whether the selected reviewers are representative. You have planned to analyse your data on a monthly basis, it is however unclear if you plan to publish on a monthly basis; if yes, how will you handle the previous version? In the Discussion, it was mentioned that there was no consensus on what constitutes intrauterine, intrapartum and postpartum transmission. Please refer to this publication - Prakesh S Shah, Yenge Diambomba, Ganesh Acharya, Shaun K Morris, Ari Bitnun. Classification system and case definition for SARS-CoV-2 infection in pregnant women, fetuses, and neonates. Acta Obstet Gynecol Scand. 2020 May;99(5):565-568. doi: 10.1111/aogs.13870.
--	--

VERSION 1 – AUTHOR RESPONSE

Response to reviewer comments

Reviewer #1

4. It is an urgent task to elucidate the effects of coronavirus infection on pregnant and postpartum women while the end of coronavirus infection cannot be seen. The studies planned by the authors are clear and sound in purpose and method and beneficial to many clinicians. Reviewer believe that this work should be carried out promptly.

We thank the reviewer for this positive comment.

Reviewer #2

5. why don't you include study which is registered in clinical trial as well. Some of the study results there is not published.

We agree with the reviewer and have already indicated that we will review registries of randomised controlled trials to detect trials that should have reported results but have not done so, in order to contact corresponding authors to obtain results (Page 8, line 6-13)

6. Please include web of science database as well.

Our search strategy includes searches on the World Health Organisations (WHO) Database of publications on COVID-19, which includes the web of science database. Our complete search strategy will be published with the main article.

7. Suggest to register the study in US clinical trial

Our study is not a clinical trial and so registration on a clinical trial registry would not be appropriate. Our living systematic review is registered on PROSPERO with registration number CRD42020178076.

8. Suggest to do flow chart

A PRISMA flowchart on study selection will be included with our main publication.

9. Suggest to determine sample size for each objectives based on previous literature

In our submitted protocol manuscript (page 16, line 15-22) we have indicated that: "A priori assumptions on statistical power, minimal clinically relevant effect and heterogeneity between trials will be used to define maximum sample size to detect such an effect. Boundaries of statistical significance according to sample size will be defined and used to determine statistical significance for each systematic review update result."

Reviewer #3

10. Can the authors elaborate on the rationale behind the proposed schedule?

A living systematic review (LSR) is an emerging approach to updating a systematic review, and there are no set rules on the frequency of updates. Our proposed schedule was a pragmatic decision based on the volume of studies we anticipated finding in our search. We are able to adapt the LSR schedule if the volume and relevance of studies being published changes, and following advice from our independent steering committee.

11. For the monthly updates, is there a cut off date?

We have no specified cut off dates, but we plan to do so at least for the next year, and the frequency afterwards will be guided by the steering committee.

12. Are you reporting on analysis from 2 weeks before?

Yes, the analyses are reported from two weeks before.

13. Will there be a final publication?

A manuscript addressing some of the questions in the protocol has been accepted by the BMJ. We will update this publication as new evidence emerges in line with the requirement of the journal over the next 2 years. The BMJ have recently published two other living systematic reviews as well as a commentary on living systematic reviews, and their frequency of publication updates <https://www.bmj.com/content/370/bmj.m2925>.

14. I believe the aim to evaluate mother-child transmission is an ambitious one as most data provide insufficient evidence for in utero, intra and peripartum mother-child transmission. Most publications have focused on early neonatal SARS-CoV-2 infection. You may wish to revise this aim to allow some flexibility.

We agree that this is an ambitious aim, and we are well placed to deliver this. We are currently working with the World Health Organization and relevant authorities to develop a standardised classification for mother to child transmission of the SARS-CoV-2 virus. We are in a unique position to evaluate the available evidence and provide the necessary recommendations on what is needed to adequately address this critical issue.

15. Will you include preprint papers or published papers only?

We include both preprint and published papers identified from a number of databases as mentioned in the literature search section of the protocol (page 12, line 5-22). All articles including preprints, will undergo the same rigorous quality assessment to determine risk of bias of the study.

16. What does "our search will be updated weekly" mean? Does it mean you will run the search weekly?

Yes, we update our searches every Tuesday to identify any new publication from the previous week.

17. It is unclear how you will include nonpregnant cases with SARS-CoV-2 infection. Please explain how you will capture such data systematically.

Comparative cohort studies identified in our weekly searches that include pregnant and non-pregnant cases are included. Data on these comparative groups are extracted in the same way as pregnant cases.

18. Table 1: Risk factors –

a. consider adding respiratory disease, cardiovascular disease, immunosuppression, smoking.

Since the submission of our protocol, we have added respiratory disease, cardiovascular disease and smoking to our list of risk factors. We will also add immunosuppression as recommended by the reviewer.

b. Parity, gestational age, mode of delivery, pregnancy status should not be considered as risk factors.

It is unclear why the reviewer states that parity, gestational age, mode of delivery and pregnancy status should not be considered as risk factors. These are all accepted risk factors for increased risk of various adverse pregnancy outcomes and it is important to assess whether they are also risk factors for COVID-19 infection or maternal and perinatal complications, in pregnant and postpartum women.

c. Outcomes - consider adding thromboembolism

We have added thromboembolism to our outcomes in table 1.

d. Offspring outcomes - there was a recent case of neonatal white matter injury; will you have "others"?

We acknowledge the recent case of white matter injury in a neonate, but our focus is on clinical complications rather than neuro-radiological findings in the offspring. Being a living protocol, we are constantly assessing the need for including any relevant outcomes after discussion with members of our steering committee.

19. Regarding duplicates, it is good that you have briefly mentioned that you have a plan to address this issue. Please be mindful that the same cases can be published by obstetricians, neonatologists, radiologists, intensivists, pathologists, admitting physicians, receiving physicians, etc. This is a potential problem that would either over- or under-estimate the effect of SARS-CoV-2 infection on maternal and perinatal outcomes.

We agree with the reviewer on the issue of the same cases being reported in multiple publications, and have put in place a robust process of identifying such instances. For each study where there is suspicion of duplication of cases, we record the country and hospital of the subjects, and cross check the timeframes of patient recruitment to identify overlaps in dates. If there is an overlap in dates, we include the largest cohort with the most relevant data. However, if the duplicate cohorts report on different risk factors or outcomes, we will include both studies for their separate data. We have decided to err on the side of caution and exclude any study where we could not rule out potential overlap in the study populations.

b. You have also mentioned that you plan to have a semi-automated process to identify these duplicates. It would be good to know more about this semi-automated process as supplementary materials.

We will report our process of identifying duplicates in the methods section of our manuscript, and report all studies that we identified or excluded as potential duplicates in the supplementary materials.

20. Please explain how you will handle missing data. I believe this could also be a significant issue, especially in regard to the earlier data. Such data are important as they represent the impact of the disease during the beginning of the outbreak. I also assume that the earlier data would be of low quality.

Missing data is an issue with all systematic reviews and meta-analysis, and unlike primary studies cannot be addressed by multiple imputation or intention to treat analysis. We believe our review is less likely to suffer from publication bias because of the frequency and speed of publication of covid related articles by journals and pre-print servers. As mentioned in point 12, all articles will undergo a rigorous quality assessment to determine risk of bias of the study.

21. One of the objectives is to determine if the rates of prevalence, clinical manifestations, outcomes and risk factors vary by: a) screening and testing strategy for SARS-CoV-2, b) selection of populations, c) risk status of the included women, timing of exposure (first, second, third trimester, postpartum), World Bank economic region (low, middle and high income) and quality of studies (low, high). In your pre-planned subgroup analysis, quality of studies was not included.

We thank the reviewer for the comment. This was omitted in error in the manuscript and has now been added to the list of sub-group analysis in page 16 line 1.

22. with your objective / pre-planned subgroup analysis, there was an item called maternal risk status. What does this item mean?

Studies defined as high risk group will include only women with pre-existing medical or obstetric risk factors, low risk if women did not have any risk factors, and any risk if all types of women were included.

23. Please describe the process of establishing a pool of peer reviewers. It would be good to know whether the selected reviewers are representative.

Our pool of reviewers was identified to provide rapid assessment of our findings before publication on our dedicated website. We identified reviewers from high as well as low-and middle-income countries, across different specialties of academics, clinical and laboratory scientists as well as methodologists. Any manuscript submitted for publication will be reviewed by independent reviewers identified by the journal.

24. You have planned to analyse your data on a monthly basis, it is however unclear if you plan to publish on a monthly basis; if yes, how will you handle the previous version?

We thank the reviewer for their comment. Following guidance by the project steering committee, we have agreed to reduce the frequency of our analysis to every 6-8 weeks for pragmatic reasons, following the publication of our initial findings. Thus, we will aim to publish a summary of our results online on our dedicated website every 6-8 weeks: <https://www.birmingham.ac.uk/research/who-collaborating-centre/pregcov/index.aspx>. All previous versions will be archived and made available for download.

Publication of living systematic reviews is a fairly new concept, with journals adopting a different stances on the frequency of updates. We will negotiate publication agreements with journals on a case by case basis. Our manuscript accepted by the BMJ will be updated three times over the next year, with a permanent link to previous versions available online.

25. In the Discussion, it was mentioned that there was no consensus on what constitutes intrauterine, intrapartum and postpartum transmission. Please refer to this publication - Prakesh S Shah, Yenge Diambomba, Ganesh Acharya, Shaun K Morris, Ari Bitnun. Classification system and case definition for SARS-CoV-2 infection in pregnant women, fetuses, and neonates. *Acta Obstet Gynecol Scand*. 2020 May;99(5):565-568. doi: 10.1111/aogs.13870.

We are aware of the publication cited by the reviewer, and there is also another case definition proposed by Blumberg et al. (Blumberg DA, Underwood MA, Hedriana HL, Lakshminrusimha S. Vertical Transmission of SARS-CoV-2: What is the Optimal Definition?. *Am J Perinatol*. 2020;37(8):769-772. doi:10.1055/s-0040-1712457). Both publications and their resulting classification systems were not reached through a formal consensus process, and there still remains questions on their suitability for defining vertical transmission of SARS-CoV-2. As mentioned in our response to 11 above, we are working with the World Health Organisation to develop a standardised classification for mother to child transmission of the SARS-CoV-2 virus, which we will be able to apply to studies identified in our LSR.